# Diagnosis of Schizophrenia Using Multimodal Data and Classification Using the EEGNet Framework

**DOI:** 10.3390/diagnostics15233081

**Published:** 2025-12-03

**Authors:** Nandini Manickam, Vijayakumar Ponnusamy, Arul Saravanan

**Affiliations:** 1Department of Electronics and Communication Engineering, School of Electrical and Electronics Engineering, SRM Institute of Science and Technology, Kattankulathur, Chengalpattu 603203, Tamilnadu, India; nm6075@srmist.edu.in; 2Department of Psychiatry, SRM Medical College Hospital and Research Centre, Kattankulathur, Chengalpattu 603203, Tamilnadu, India; arulsarr@srmist.edu

**Keywords:** audio and video signals, electroencephalogram (EEG), machine learning (ML), multimodal data, schizophrenia

## Abstract

**Background/Objectives**: In recent years, people have been facing a lot of difficulties in handling stress, emotions, social, and behavioral issues, which have led to severe mental disorders. Schizophrenia is one disorder that requires more attention. This disorder is characterized by positive or psychotic symptoms, negative symptoms, and cognitive symptoms, which makes diagnosis and treatment complicated. The main objective is to identify the degree of severity of symptoms through multimodal data and classify them using the EEGNet framework. **Methods**: Multimodal data are collected. To identify the severity of symptoms of schizophrenia, initial screening is performed through assessment tools such as the Positive and Negative Symptoms Scale (PANSS), Brief Negative Symptom Scale (BNSS), Scale for Assessment of Negative Symptoms (NSA-16), and Scale for Assessment of Positive Symptoms (SAPS). Designed photo elicitation and VR box video stimuli are used for data collection. The patients are asked to express their thoughts upon viewing photos shown through a photo elicitation task. The patients are given Virtual Reality (VR) stimuli where videos will be played in a VR box and patients are asked to express their thoughts. Patients’ facial expressions and speech signals are captured through webcam while performing these tasks. Finally, the electrical activities of the patients are assessed through a 14-channel EEG headset. A novel method of fusing and embedding of normalized multimodality features into the EEGNet architecture is carried out that enables combined utilization of electrophysiological information from EEG and complementary behavioral–affective cues from other modalities, thereby enhancing classification performance while retaining the architectural efficiency of EEGNet. **Results:** The reliability and validity of the questionnaire are statistically analyzed and found to be α = 0.761. The sum of variance of PANSS is about 27.08, SAPS is about 28.61, and BNSS is about 29.92 with *p* < 0.05. This EEGNet model displays an accuracy of 0.99, precision of 0.99, recall of 0.98, and F1-score of 0.99 for healthy and a precision of 0.98, recall of 0.99, and F1-score of 0.99 for schizophrenia-affected patients and ROC AUC of about 0.9989. **Conclusions**: This system proves to be a promising method for the diagnosis of schizophrenia and thereby enhances the performance of the system.

## 1. Introduction

Schizophrenia is a serious psychiatric disorder that affects the thoughts, speech, behavior, emotions, physical, and social well-being of a person. Schizophrenia affects around 24 million (0.32%) people worldwide according to the World Health Organization (WHO). According to the National Mental Health Survey (NMHS) in India, the occurrence and prevalence of schizophrenia spectrum disorders were examined during 2015 and 2016 through a multistage, stratified, random cluster sampling technique [1]. It was found that the prevalence of schizophrenia spectrum disorder was about 1.41% in their entire life, and nearly 0.42% people were diagnosed with schizophrenia at that moment. Schizophrenia is characterized by impairments in the way things are perceived and changes in behavior. This causes psychosis, which may affect the personal, family, social, educational, and occupational functioning of a person since they survive in a world that is disconnected from reality. Participants who show at least one of the symptoms of delusions, hallucinations, or disorganized speech during a period of one month are included in this study. Further, this disorder is diagnosed between late teens and early thirties. In males, it is found earlier between late adolescence and early twenties and, similarly, in females, it is found between early twenties and early thirties. Participants who show a history of autism spectrum disorder, communication disorder in childhood, or depressive or bipolar disorder are excluded from this study. This disorder is more commonly found in men than in women. Schizophrenia shows a variety of symptoms, which makes diagnosis and treatment very difficult. It becomes hard to recognize a person with schizophrenia. Experts view schizophrenia as a spectrum of conditions.

Signs and symptoms are classified based on the characteristics of the brain’s cognitive functions. The first kind of symptom comprises positive or psychotic symptoms. The positive symptoms are characterized by the following:Delusion: The affected person has false beliefs and is strongly convinced of them despite contradictory evidence.Hallucinations: The affected people can hear unusual voices or sounds.Disorganized or incoherent speech: People might have problems conveying their thoughts. They find difficulty in organizing their thoughts while speaking. People find it difficult to understand.Grandiosity: The person feels superior to others.Suspicion: The person frequently feels scared, and becomes suspicious of everything around him/her.

The negative symptoms are characterized by emotional withdrawal, disinterest in their appearance or personal hygiene, poor social involvement, lack of spontaneity, and difficulty in abstract thinking. Apart from these positive and negative symptoms, they also show cognitive symptoms characterized by a lack of memory, poor attention, low processing speed, a lack of visual and verbal learning, substantial deficits in reasoning, a lack of planning, and problem-solving. Integration of genetic predisposition along with environmental, social, and psychological factors causes abnormalities in neurodevelopmental features, and this leads to brain dysfunction and improper balance of chemicals, which results in schizophrenia. Analyzing these factors and symptoms through machine learning helps to enhance the performance in diagnosing schizophrenia.

EEGNet is the most efficient and compact convolutional neural network (CNN) architecture, mainly suitable for EEG-based brain–computer interface (BCI) tasks. This method is highly effective in the detection of schizophrenia [2]. This architecture mainly extracts spatio-temporal and frequency-based features from EEG data. The diagnosis process is carried out by extracting physiological data, either EEG/ECG or MRI data, textual data through questionnaires, video signals through photo elicitation tasks, and speech signals through VR stimuli assessment tasks from the control group and schizophrenia patients. These raw data undergo pre-processing, where the artifacts and noises are removed before being fed into a federated learning network. Applying machine learning algorithms, the features and patterns are trained in a model to classify the presence of schizophrenia. This paper mainly concentrates on the questionnaire and how the system behaves when the publicly available dataset is fed into the proposed EEGNet architecture.

## 2. Related Work

This article focuses on the use of EEGNet in the diagnosis of schizophrenia by statistically analyzing self-report questionnaires and studying the traits of schizophrenia using textual and physiological data aspects. Various data privacy preservation techniques are also examined, which improve the system’s functionality.

For the diagnosis of schizophrenia, a prediction model is developed using aperiodic neural activity, which has been linked to the excitation−inhibition balance and neural spikes in the brain [3]. A questionnaire and resting-state electroencephalogram (EEG) are used to investigate the link between anomalies in the first episode of schizophrenia spectrum psychosis (FESSP) and to study the intensity of symptoms. The severity of symptoms is assessed through assessment tools such as the Brief Psychiatric Rating Scale (BPRS), the Scale for Assessment of Negative Symptoms (SANS), and the Scale for Assessment of Positive Symptoms (SAPS). The EEG data were collected using a 60-channel cap based on the 10–20 international system, and the power spectral density of each electrode was measured for every individual. A correlation of r = 0.89 and *p* < 0.001 is displayed in the aperiodic offset and exponent features.

Several studies have explored the characteristics and benefits of multimodal fusion, which is used to capture complex emotional and behavioral patterns that are associated with mental illness [4]. A study of mental health videos on the Douyin platform illustrates how different modalities like visual, linguistic, and textual cues help in understanding how people think about and talk about mental illness.

Detection of schizophrenia is improved by using two new methods, namely Automatic Speech Recognition (ASR) and Natural Language Processing (NLP) [5]. Word Error Rate (WER) is used to measure the ASR data. Similarly, error type and position are used to examine the speech’s character qualitatively. The accuracy of classification is improved by measuring the word similarity. To investigate the similarity between automatic and manual transcripts, two random forest classifiers were used. The accuracy of manual transcription was 79.8% whereas that of automatic transcription was found to be 76.7%. The accuracy of classification was improved when NLP and ASR were combined.

An automatic EEG detection method for the early diagnosis of schizophrenia is designed using a vision transformer [6]. 19 channels of electrode placements are used in a 10–20 international system to map 1D EEG sequence data into a 3D image. This preserves the temporal and spatial characteristics. The performance of the model is evaluated through leave-one-subject-out cross-validation and 10-fold cross-validation methods. This was examined for the assessment of both subject-independent and subject-dependent categories. For the subject-independent category, the model’s average accuracy was 98.99% and the model’s average accuracy was 85.04% for the subject-dependent category. This supports early diagnosis and treatment.

Multimodal MRI and deep graph neural networks are used for extracting neuroimaging modalities to assess the presence of schizophrenia [7]. This framework reveals region-specific weight irregularities that are linked to transcriptomic features. Thus, integration of neurobiological and behavioral data has proved as a promising technique in understanding the heterogeneity of disorder. Similarly, machine learning models integrated with neuroimaging modalities are used to differentiate schizophrenia, bipolar disorder, and borderline personality disorders by capturing multimodal features in order to improve the diagnostic accuracy.

An enhanced system with a single-channel EEG can be used to diagnose schizophrenia automatically with little input data [8]. To create a model, this EEG data is extracted using knowledge distillation and transfer learning. The effectiveness of the system can be improved using this method, with the help of the continuous wavelet transform (CWT), which makes use of previously learned models, and EEG data are also converted to images. The automatic diagnosis of schizophrenia can be improved by expanding the number of pre-trained models in the image domain. The accuracy is improved by 97.81% by integrating self-distillation and VGG16 in the P4 channel.

To extract the spatial and temporal patterns from the EEG, temporal and spatial convolutions are used. A novel deep learning method called CALSczNet is applied to improve the detection performance [9]. The technique uses local attention (LA), temporal attention (TA), and long-short-term memory (LSTM) to address the discriminative characteristics of non-stationary EEG signals. Using the publicly available Kaggle dataset for simple sensory tasks, the model’s performance can be assessed through 10-fold cross-validation. The model has achieved about 98.6% accuracy, 98.65% sensitivity, 98.72% specificity, and a 98.65% F1-score. The model is capable of detecting schizophrenia at an early stage.

Resting-state functional magnetic resonance imaging (rs-fMRI) plays an important role in analyzing the brain’s functional connectivity for diagnosing schizophrenia at an early stage [10]. To determine the association of brain connectivity data, graph attention networks (GAT) are utilized to derive node-level features. The integration of the multi-graph attention network (MGAT) and a bilinear convolution (BC) neural network makes measuring functional connectivity easier. Grid search and multistage cross-validation are used in the MGAT-BC model to assess performance. This approach improves the diagnosis of schizophrenia by capturing topological aspects with an accuracy of 92%.

The overlap of symptoms of schizophrenia with other diseases can be resolved by a data-driven diagnosis that uses EEG data to identify brain connection biomarkers [11]. By extracting the EEG spectral power data, a new feature interaction-based explainability technique for multimodal explanations is created. To find the biomarkers for neuropsychiatric illnesses, the impacts of schizophrenia are studied on various frequency bands, the alpha, beta, and theta frequency bands, and the explainable machine learning models. The mean and standard deviation of relevance for each fold were used to evaluate the model’s performance. For the mean, this model has shown an accuracy of 0.73 and an F1-score of 0.70. Similarly, for standard deviation, this model has achieved an accuracy of 0.09 and an F1-score of 0.13.

The nature of psychotic episodes and the fact that symptoms vary from person to person have made it difficult to identify them [12]. Important information may occasionally be omitted when information is extracted from electronic health records (EHR). To diagnose psychosis, admission notes are extracted using Natural Language Processing (NLP) techniques. The models are trained using keywords, and the algorithms are assessed through a rule-based methodology. The effectiveness of the two models’ keyword extraction was measured. The first model achieved better results using an XGBoost classifier that utilizes term frequency-inverse document frequency (TF-IDF) for extracting features from notes through expert-curated keywords and achieved an F1-score of 0.8881. Another model achieved an F1-score of 0.8841 using BlueBERT to extract the same notes.

Recent studies have focused on the diagnosis of schizophrenia and analyzing the severity of symptoms using multimodal data. Self-supervised speech representations are used to assess the symptoms of schizophrenia [13]. Multimodal biomarkers are used for measuring the severity of individual symptoms. These biomarkers are used for capturing speech and facial signals in psychiatric assessment. The effectiveness of speech, language, and orofacial signals is studied for remotely assessing the positive, negative, and cognitive symptoms.

P300 is utilized to examine the processing, stimulus-based responses and the instability of brain disorders [14]. When tracking the locations of the brain during the cognitive process, more emphasis is paid to trial-to-trial variability, or TTV. This TTV is in a time-varying EEG network across the beta1, beta2, alpha, theta, and delta bands and is examined in this paper for the diagnosis of schizophrenia. A cross-band time-varying network can successfully extract the characteristics of schizophrenia with an accuracy of 83.39%, sensitivity of 89.22%, and specificity of 74.55%, according to the performance evaluation.

Social media-based multimodal data is also used to identify the early signs of mental distress, which highlights the role of digital behavioral footprints in screening [15]. Digital footprints represent the participant’s status, comments, likes, language styles, sleep–wake patterns, and social activity, which demonstrates cognitive and emotional fluctuations. These modalities are analyzed using multimodal machine learning models and their performance is evaluated.

Schizophrenia patients experience disturbances in emotion, expression, and social interaction, and conventional methods depend upon in-person visits. This can be resolved by developing a remote, multimodal digital system that helps physicians assess the emotional dynamics in remote areas [16]. This method uses a dialog-based human–machine interaction and multimodal markers that can detect and track the emotional changes in schizophrenia patients.

Anxiety symptoms in social anxiety disorder (SAD) are predicted using machine learning algorithms through multimodal data from Virtual Reality (VR) sessions [17]. The severity of various anxiety symptoms is assessed through questionnaires and six VR sessions where the participants are allowed to interact with virtual characters and perform meditation-based relaxation exercises. Participants are exposed to video recording and physiological data during the VR session. The Extended Geneva minimalistic acoustic parameter set (eGeMAPS) is used to extract the acoustic features through the open SMILE toolkit. Core symptoms and cognitive symptoms of SAD were evaluated through catBoost and XGBoost models.

Digital assessment and monitoring of schizophrenia patients can be performed remotely by developing a scalable multimodal dialog platform [18]. This system employs a virtual agent to carry out automated tasks using the Neurological and Mental Health Screening Instrument (NEMSI), a multimodal dialog system. Twenty-four participants diagnosed with schizophrenia were assessed through PANSS, BNSS, CDSS, CGI-S, AIMS, and BARS scales, speech signals were extracted through PRAAT, and facial expressions were extracted through MediaPipe and the FaceMesh algorithm. Speech metrics showed a result of mean AUC: 0.84 ± 0.02, which performed better than facial metrics with mean AUC: 0.75 ± 0.04 in classifying patients from controls. Speech and facial metrics showed test–retest-reliability performance of about >0.6 (moderate to high).

Communicative and expressive features that distinguish schizophrenia patients from healthy controls are evaluated by developing a multimodal assessment model using machine learning [19]. Thirty-two participants with schizophrenia were assessed on the assessment battery for communication (ABaCo) for inter-rater reliability. Linguistic and extralinguistic scales assess the communicative and gestural. This assessment consists of 72 items in the form of interview interactions, which were recorded for about 100 short clips with a duration of 20–25 s each. The decision tree model was trained and achieved a mean accuracy of 82%, sensitivity of 76%, and precision of 91%.

Early diagnosis of mental health among corporate professionals can be assessed by developing an emotion-aware ensemble learning (EAEL) framework [20]. This framework assesses facial expression and typing patterns. Facial expressions were captured through webcam interaction and cognitive and motor processes through typing patterns. SVM, CNN, and RF algorithms are used in this framework to capture emotions like sentiments, happiness, sadness, anger, surprise, disgust, fear, neutrality, confusion, and typing tasks that measure average typing speed, key hold time, key latency time, number of typos, and stress level. The EAEL framework showed an accuracy of 0.95, precision of 0.96, recall of 0.94, and F1-score of 0.95.

To analyze schizotypy traits across the psychosis spectrum, an uncertainty-aware model is developed for extracting acoustic and linguistic features [21]. From 114 participants taken to the study by the author of [21], 32 individuals with early psychosis and 82 individuals with low or high schizotypy were identified. The performance of speech and language models in classification and multimodal fusion models is analyzed. The multidimensional schizotypy scale (MSS) and Oxford-Liverpool Inventory of Feelings and Experiences (O-LIFE) were used for initial level screening. Acoustic features were extracted using open SMILE and whisperX. The model performs well across various interactions when employing random forest, linear discriminant analysis, and support vector machine SHAP for model prediction, achieving an F1-score of 83% and ECE 4.5 × 10^−2^.

Acoustic speech features are extracted for distinguishing schizophrenia spectrum disorders (SSD) from healthy controls [22]. The speech markers are used to recognize the negative and cognitive symptoms. Participants are allowed to take up a speech-eliciting task where they are asked to express their thoughts on viewing the picture. Prosody, temporal, spectral, articulatory, and rhythm dimensions are some of the parameters that are assessed using ML. Automated and objective architecture is also used for the detection of schizophrenia using multimodal behavioral features that fuses feature-level and decision-level features, which is responsible for identifying which modality is responsible for the symptoms obtained. [23].

This work mainly focuses on the role of EEGNet in diagnosing schizophrenia through questionnaires and EEG data. The questionnaire data are analyzed through statistical approaches and the EEG data are analyzed by pre-processing the signals before feeding them into the EEGNet network. Machine learning techniques are used to extract spatio-temporal and frequency-based data and find patterns in order to distinguish between patients and those with schizophrenia. The methods of preserving data privacy, thereby enhancing the performance of the system, are also analyzed.

## 3. Materials and Methods

### 3.1. Data Collection and Preparation

The participants’ data are collected from SRM Medical College Hospital and Research Centre, Chengalpattu, Chennai. The Institutional Ethics Committee (IEC) had approved data collection through a formal ethical clearance meeting, which was issued on 23 July 2024 with Register Number (REG.No:ECR/8848/INST/TN/2013/RR-19). The participants were informed about the study, and their participation was purely voluntary. Participants were free to leave the study at any time. Before starting the study, consent forms were duly signed by the participants. Initially, questionnaire data were taken from participants through online and offline modes. These questionnaires are framed to analyze the psychotic or positive, negative, and cognitive symptoms of schizophrenia. Positive and Negative Symptoms Scale (PANSS), Brief Negative Symptom Scale (BNSS), Scale for Assessment of Negative Symptoms (NSA-16), and Scale for Assessment of Positive Symptoms (SAPS) assessment tools are taken from inventories that follow the guidelines framed by the Diagnostic and Statistical Manual of Mental Disorders *(*DSM-5) and International Classification of Diseases (ICD-11) for diagnosis. These three questionnaires were sent to healthy controls through Google Forms, and for patients, the questionnaires were explained directly in the presence of attenders, and their responses were entered by the investigator. There were 20 responses received online, consisting of 10 male and 10 female participants, with age groups ranging from 20 to 50 years. Using Likert scaling and data coding, the responses are converted into values, and a threshold score is set as a total score of ≥60. Based on the scores obtained, the severity of symptoms is classified as moderate (50–60), mild (40–51), less (30–41), and very less (20–31). Similarly, there were 20 responses received from the hospital, consisting of 8 male and 12 female patients with an age group ranging from 17 to 58 years, with a threshold score set as total score ≥50. Based on the scores obtained, the severity of symptoms is classified as very moderate (50–60), moderate (60–70), high (70–80), very high (80–90), and extreme for scores >90. These values were statistically analyzed through Analysis of Variance (ANOVA), T-test, and chi-square methods. Then, the severity of data is compared with the statistical methods and control groups and schizophrenia patients are categorized in each assessment tool. This evaluates the reliability and validity of the data. The characteristics of symptoms that are more dominant in a person are analyzed. Figure 1a,b depict the sample questionnaire data collected in the hospital.

Secondly, the participants are assessed on the severity of symptoms through a photo elicitation task. This session is a semi-structured autobiographical interview that takes around 10 min, where nearly 15 photos related to the symptoms of schizophrenia, like hallucinations and delusions, are shown, and the participants are asked to express their feelings or describe each photo displayed to them. This task is captured through a webcam to analyze their facial expressions and speech signals. Video analytics in terms of frames-per-second and resolution are analyzed. Emotions like happiness, sadness, anger, surprise, disgust, fear, neutrality, and confusion are evaluated from the facial expressions captured. In facial expressions, average velocity, acceleration, and jerk for various actions across facial parts are observed. Acoustic features like frequency, spectral, relative energy in different frequencies, pitch, and jitter are extracted from the captured data. Figure 2 depicts the sample photo elicitation task collected from patients in the hospital.

Thirdly, the participants are assessed on the severity of symptoms through a Virtual Reality (VR) assessment stimulus task. This is a pre-assessment test for checking the compatibility and how long participants can wear the VR box during the session. In this session, a video that consists of scenarios related to the symptoms will be played in the VR box for about 10 min, and participants are asked to express their feelings after each scenario. The patients get stimulated by a few scenarios that connect with their symptoms and start expressing their view and what kind of similar scenario they faced. Two patients refused to take up this task, and one patient could not take up the task since he had hearing issues. Three patients felt uncomfortable wearing the VR box and viewing the video in it. This task is captured through a webcam to analyze the speech signals. The sample VR assessment taken by the patient in the hospital is depicted in Figure 3.

Physiological data are used to monitor brain activities at different frequency bands at a particular sampling rate. EEG is a device that measures the electrical signals when electrodes are placed at different scalp positions. A 16-channel EEG hardware from RMS, EEG-32 Recorders and Medicare Systems, Chennai, India captures the activities concentrating on the brain regions responsible for cognitive functions. This follows a 10–20 international system layout for electrode placements that covers the frontal, parietal, temporal, and occipital regions. The data are collected under an open and rest condition of eye movements. The raw EEG data signals undergo a pre-processing stage where the noise present in the electrical signals is removed by normalizing each EEG channel to the Z-score. Data augmentation is performed using 2-s windows with overlapping. These pre-processed signals are fed into an EEGNet network. This architecture is a lightweight CNN that is designed specifically for EEG signals. This system helps in generalization and model robustness. This consists of very few parameters and uses depth-wise separable convolutions for spatial filtering, separable convolutions for frequency filtering, and dropout and batch normalization for regularization and stability. Using ML algorithms, features are extracted, and performance metrics like accuracy, precision, recall, and F1-score are measured for healthy and affected patients. Figure 4 displays the structure of the proposed system.

The existing method has taken datasets from the publicly available repository [24]. The dataset includes 14 healthy controls and 14 patients with schizophrenia. The electrical signals are stored in European Data Format (EDF), where the raw data are extracted at 250 Hz along 19 EEG channels. The duration of the data collection is about fifteen minutes. The electrodes were placed at Fp2, F8, T4, T6, O2, Fp1, F7, T3, T5, O1, F4, C4, P4, F3, C3, P3, Fz, Cz, and Pz positions, and their signals for 28 participants were recorded. The patients with severe neurological disorders were excluded from the study. These EEG signals are pre-processed before being fed into a neural network. A total of 1142 EEG segments were employed, out of which each segment consists of 6250 × 19 sample points. To make deep learning model training easier, normalization is utilized to scale the signals to a standard range of values.

An eleven-layered deep CNN model is developed to distinguish between healthy controls and schizophrenia patients for subject-based and non-subject-based testing [25]. Different methods are implemented in the CNN architecture for subject-based and non-subject-based testing. The average pooling layer is used for feature extraction in subject-based testing, whereas global average pooling layers are used for generalized predictions. To minimize the loss function during the training process and to classify the model based on subjects and extract significant features, Adam optimization and Leaky Rectified Linear Unit (LeakyReLU) were implemented. Cross-validation was performed using a 14-fold, and the models were trained for 50 epochs. Similarly, for non-subject-based testing, feature extraction is performed using a max pooling layer. Here, cross-validation is performed using a 10-fold, and the models were trained for 70 epochs. Performance metrics like accuracy, sensitivity, specificity, and positive predictive values (PPV) were measured for every fold. Classification results per fold for subject-based testing and non-subject-based testing showed that non-subject-based testing performed better than subject-based testing. The performance of non-subject-based testing showed about 98.07% accuracy, 97.32% sensitivity, 98.17% specificity, and 98.45% PPV. Based on the confusion matrix, it was inferred that 13.18% of healthy controls were miscategorized as schizophrenia patients and 23.32% of healthy controls were misclassified as schizophrenia patients in non-subject-based testing.

### 3.2. Proposed EEGNet-Based Diagnosis of Schizophrenia

This article contributed by embedding the multimodality fussed features into the EEGNet architecture. We have developed normalized feature set embeddings from four modalities: (1) normalized questionnaire scores, (2) normalized speech features extracted by giving Photo Elicitation and VR Assessment Stimuli, (3) video emotion extracted features by giving Photo Elicitation and VR Assessment Stimuli, and (4) EEG data features. Those features were concatenated to form a joint feature vector.F_fusion_ = [F_EEG_∥F_speech_∥F_video_∥F_q_].

This integration of this multimodal fused feature enables EEGNet to jointly exploit electrophysiological information from EEG and complementary behavioral–affective cues, thereby enhancing classification performance while retaining the architectural efficiency of EEGNet.

EEGNet is an efficient and simple CNN architecture that is suitable for EEG-based brain–computer interface (BCI) tasks and effectively diagnoses the presence of schizophrenia disorders. This architecture consists of a temporal convolution layer, a depth-wise spatial convolution layer, a separable convolution layer, and a pooling and dropout layer. EEGNet model samples are collected at a 128 Hz sampling rate with ‘C’ channels and ‘T’ time samples. The network starts with a temporal convolutional layer that performs 2 convolutional steps in sequence (Conv2D), capturing EEG frequency bands (delta, theta, alpha, and beta) and operating across the time dimension. This layer is equivalent to band-pass filtering. The next layer, depth-wise spatial convolution, learns spatial filters per frequency that mimic Common Spatial Pattern (CSP) and operate across EEG channels. A separable convolution layer combines spatial and temporal features that separate the depth-wise and pointwise convolution for efficiency. The last layer is the pooling and dropout layer that reduces overfitting and controls model complexity.

Depth-wise separable convolution comprises two components: depth-wise convolution and pointwise convolution. Depth-wise convolution applies a single filter per input channel, and pointwise convolution combines outputs from the depth-wise step to new features. EEGNet uses depth-wise convolution across channels, extracting the spatial features. This is similar to the Common Spatial Patterns (CSP) method of EEG analysis. This method is suitable for smaller EEG datasets. In separable convolution, the model first applies filtering over time that captures frequency dynamics and then performs filtering across channels. Since EEG is a time-series data where different frequency bands (delta, beta, alpha, theta) have diagnostic value, this layer allows the model to differentiate between frequency bands effectively. Dropout is used to prevent overfitting issues during training. This is achieved by making the model learn redundant and generalizable patterns, thereby increasing robustness and generalization. Since EEG signals vary by subject, session, and channel, batch normalization helps in reducing variability and improving performance. Figure 5 depicts the EEGNet architecture.

The existing CNN architecture consists of 9 layers of Convolution layers and max pooling layers below each convolution layer. This structure is developed with the Leaky Rectifier Linear Unit (LeakyRelu) with a dropout rate of 0.5 and different rates at different layers. Feature extraction is performed through the max pooling layer, and significant features are extracted through the global average pooling layer, which consists of a dense layer with a softmax activation function. The global model’s performance in diagnosing schizophrenia is evaluated by calculating accuracy, precision, recall, F1-score, and Receiver Operating Characteristics (ROC) metrics. The performance of the proposed and existing models is compared, and results show that the proposed framework performs better than the existing method, with an accuracy of 99.25, a precision of 90.79, a recall of 96.14, and an ROC of 99.73. A CNN-transformer fusion model is developed that extracts EEG signals through motor imager, P300, and steady-state visual evoked potential (SSVEP) tasks [26]. This architecture provides a lightweight, deep learning model that works across different experimental setups without extensive tuning and combines temporal, spatial, and depth-wise separable convolution that makes the system efficient. The automatic detection of brain abnormalities through EEG patterns is also evaluated using 5 CNN architectures, namely Attention-based, ResNet50, Inception-V4, efficientnetBo, and squeeze and excitation net [27]. On analyzing the performance of diagnosis, attention-based CNN achieved a training accuracy of 99.39% and validation accuracy of 98.95%. There is a need for biomarkers that solely support the diagnosis of major depressive disorder (MDD) and bipolar disorder (BD) using the EEG modality [28]. In an EEG-based study, the diagnosis is explored using different shallow and deep neural networks. This paper analyzes experimental protocols using EEG biomarkers, extract their features, and explores different model types for the detection of MDD and BD. For MDD, various EEG experiments like resting-state vs. task, number of channels, electrode locations, biomarkers, neural network architectures, and their performance metrics are analyzed. Similarly, for BD, fewer studies that cover the protocols, neural network models, and their features and performance are analyzed. There is a need for the development of biomarkers and automatic systems that can support early diagnosis of depression and also forecast the risk of relapse [29]. This can be investigated and evaluated through three modalities, such as self-questionnaires, audio–visual cues, and EEG signals. This literature mainly focuses on how machine learning is used to recognize the clinical and non-clinical symptoms of depression. Various datasets, experimental data, methods, architectures, and evaluation metrics are explored. Multimodal approaches are used to extract speech features through Praat (v6.2.17) software, video features are extracted using media pipe face mesh, and text features are extracted through AWS transcribe using Python package spaCy, version 3.5.3 for assessing the symptoms of schizophrenia [30]. A web-based dialog system evaluates the positive, negative and cognitive symptoms using multimodal biomarkers.

There are some key features in EEGNet architecture that enhance the feature extraction, generalizability, and interpretation of schizophrenia detection. The Squeeze and Excite (SE) attention mechanism is used for improving the channel-wise feature responses, since this SE block assigns higher weights to important features from EEG channels and suppresses the rest. Thus, the most important features related to brain abnormalities are extracted, which improves interpretability. Another feature is a residual connection that is similar to ResNet architecture, which combines the outputs from the previous layers along with later ones in the neural network, making the system robust. Also, this architecture combines a 2-s overlapping window segmentation that allows detection of even the microstate change, thereby improving the real-time adaptability of the system. Thus, the proposed EEGNet architecture has shown better performance than the existing CNN-LSTM architecture in terms of architecture, input processing and segmentation, learning efficiency, and practical deployment. Table 1 shows the comparison between the proposed and existing architectures.

## 4. Results and Discussion

The statistical results obtained from the PANSS, SAPS, and BNSS scores indicate the severity of symptoms. For the healthy control group, based on the threshold, the severity levels were categorized as follows: ‘Mild, less, very less, very moderate, moderate, high, very high, and extreme’ for each assessment tool (PANSS, SAPS, and BNSS). These severity levels are compared using statistical methods, such as ANOVA, T-test, and chi-square, across all scales. On merging the total mean score and severity levels of PANSS, SAPS and BNSS, the results show that PANSS displays the highest mean score value of about 38. BNSS has a slightly lower score of around 37, and SAPS displays the lowest mean score of about 32–33. Based on the severity level distribution, nearly 40% of individuals show very less symptoms, 26.7% show less and moderate symptoms, and 6.7% individuals show mild symptoms. On applying statistical methods, ANOVA showed an F score of about 67.4797 and *p*-value of 2.674 × 10^−9^, the Welch *t*-test showed a t-score of about −8.374 and *p*-value of about 1.364 × 10^−7^, and the chi-square showed a goodness of fit (GOF) value of about 24.8. Figure 6a displays the mean score across PANSS, SAPS, and BNSS assessment tools of the control group, and (b) represents the severity distribution across three assessment tools of the healthy control group.

For the schizophrenia patients’ group, based on the threshold, the severity levels were categorized as follows: ‘Mild, less, very less, very moderate, moderate, high, very high, and extreme’ for each assessment tool (PANSS, SAPS, and BNSS). These severity levels are compared using statistical methods, such as ANOVA, T-test, and chi-square, across all scales. On merging the total mean score and severity levels of PANSS, SAPS, and BNSS, the results show that PANSS displays the highest mean score value of about 73, indicating high positive, negative, and general symptoms. SAPS has a slightly lower score of around 70 and BNSS displays the lowest mean score of about 69. Based on the severity level distribution, nearly 38.9% of individuals show high symptoms, 33.3% show moderate symptoms, 22.2% individuals show very moderate symptoms, and 5.6% individuals show very high symptoms. Upon applying statistical methods, ANOVA showed an F score of about 44.827 and a *p*-value of 4.704 × 10^−7^; T-test was not performed since they need more than one sample in both low and high groups; and the chi-square showed a goodness of fit (GOF) value of about 35.473. Figure 7a displays the mean score across PANSS, SAPS, and BNSS assessment tools of the schizophrenia patients’ group, and (b) represents the severity distribution across three assessment tools of the schizophrenia patient group.

Photo elicitation and VR stimuli task data are captured in mp4 video format. There are, in total, 20 videos for healthy controls and 20 videos for schizophrenia patients that were taken for each task. Facial expressions and speech signals are extracted through ‘PsycheScan AI’, a multimodal psychological disorder application. This application is used to recognize various facial emotions like happy, sad, angry, fear, surprise, and neutral when no expression is given and also estimates the emotion’s intensity. Different facial emotions can be detected in real-time and can be estimated through an emotion intensity radar plot. This radar chart represents the strength of each facial emotion that is detected with a scale ranging from 0 and 1. In the video segment, the number of occurrences of each emotion is also calculated. From the samples uploaded, the most dominant emotional state that occurred many times is sadness with the highest intensity of about 0.6. For a sample, it occurred around 94 times. This sample showed negative emotional pattern. Though emotions vary, sadness is the most frequent and intense emotion found in affected patients. Similarly, a sample of a healthy control was taken, and facial emotions were detected and estimated. Here, the most dominant emotion that was strongly expressed was happy with the highest intensity of about 0.8; the next highest emotion was neutral. This shows that, emotionally, a healthy person has a positive affect and shows emotional stability. A sample real-time facial expression of a schizophrenia patient and healthy control is shown in Figure 8. (a) represents the facial expression of a schizophrenia patient and (b) represents the facial expression of a healthy control.

Speech signals are obtained through a bar chart that estimates the acoustic speech features such as pitch variation, speech rate, pause frequency, intensity, jitter, and shimmer. Speech signal from the above schizophrenia sample is taken and it shows a pitch variation of about 0.8 that represents high expressive speech having a slow speech rate of about 0.25. The intensity of speech has a relatively loud voice of around 0.7 and moderate voice instability (jitter) of about 0.5 with a high vocal instability (shimmer) of about 0.9. Hence, the acoustic analysis shows that the person was expressive but had a slow and continuous speech with strong vocal intensity. Higher shimmer and moderate jitter represent the emotional strain. Similarly, the speech signal of a healthy control showed moderate intensity of about 0.42, and a very rapid speech of about 1. The intensity of speech was very loud with high vocal energy and moderate jitter of about 0.58 with a high vocal amplitude instability of about 0.70. Hence, the acoustic analysis shows that the person speaks with loud and quick frequent pauses, showing high emotional speech. Increased shimmer and jitter show high vocal strain. A sample of normalized acoustic speech features of a schizophrenia patient and healthy control is shown in Figure 9. (a) represents the acoustic speech features of a schizophrenia patient and (b) represents the acoustic speech features of a healthy control. Similarly, a comparative visualization of various acoustic features of schizophrenia patients and healthy control is depicted in (c) and (d).

Furthermore, these facial emotions and speech signals can be quantified using facial action units (FAU) and vocal feature timeline. The FAU displays the activation intensity of different facial muscle movements. There are nearly 20 facial units ranging from AU1 to AU20 and the intensity ranging from 0 to 1. Figure 10a shows the plot of measured intensity for each AU of schizophrenia patients. The AUs with high intensity that range from 0.8 to 0.9 represent muscle groups that are strongly activated, indicating either chin raise and lip compression, disgust or frustration emotion (AU17-19), and eye tightening and stress related expressions at AU7/8. Similarly, lower intensity that ranges from 0 to 0.1 shows symptoms like minimal activation with no anger, little brow lowering and cheek puffing. Similarly, (b) shows the plot of measured intensity for each AU of healthy controls. In this figure, AU10 and AU17/18/20 display near-maximum intensity of about 1 that represents strong facial expressions. AU5 and AU14 display high activation, ranging from 0.75 and 0.85, that represents eye widening and cheek movement. Here, the pattern is non-linear with multiple peaks indicating mixed emotional expressions.

The vocal feature timeline is used to quantify the speech signals that compares pitch (represented by a blue line) and intensity (represented by a green line) over a 90-s timeline. Both the signals have a rise-and-fall waveform indicating emotional fluctuations. In schizophrenia patients, there is an increasing intensity and slight drop in pitch initially from the 0 to 20 s timeline. Intensity has a sharp rise from 0 to 75 and pitch is high at 80 where the patient starts with an expressive tone. From 20 to 40 s, there is strong drop in both intensity and pitch. This period represents reduced vocal responsiveness. From 40 to 60 s, the waves show diverging dynamics where the pitch rises more steeply than intensity. From 60 to 90 s, there is a strong peak where the pitch and intensity have the highest peak of about 85 to 95. Similarly in healthy controls, both the pitch and intensity display oscillatory patterns. The waveform shows repeated rise and fall that indicates shifts between low arousal and high arousal phases. From 0 to 20 s, there is a moderate rise in pitch (60–70) and intensity (70) that indicates steady pitch, then decrease in vocal energy. From 20 to 40 s, there is a sharp decline in pitch (55–60) and intensity (10–20) indicating low expression. From 40 to 60 s, both the pitch and intensity rise sharply, indicating strong emotional burst. From 60 to 80 s, intensity again rises to another peak of about 90 and pitch drops by about 40; that indicates mixed emotions and speech transitions. From 80 to 90 s, pitch rises again and intensity drops again, indicating high tone with less loudness. Figure 11a represents the vocal feature timeline of schizophrenia patients and (b) represents the vocal feature timeline of healthy controls.

Multimodal correlation heatmap is used to find the relationship between facial action units (FAU) and speech features. This describes the interaction between facial expressions and vocal characteristics. The red color in the figure represents positive correlation that show the features’ increase together, the blue color indicates negative correlation that show the features’ movement in opposite directions, and light shades indicate weak or near zero correlation. Various AUs show strong positive correlation with speech features. AU25, AU2, AU12, and AU1 correlate with shimmer and harmonic-to-noise ratio which represent high emotional intensity. AU6, AU15, and AU25 correlate with jitter, indicating that during calm vocal states, these facial expressions exhibit strong emotions and stabilized pitch. Mixed correlation is observed with positive correlation between AU25 and AU2 and negative correlation between AU6 and AU15, indicating that upper facial movements correlate with higher pitch. Similarly, lower face tension correlates to stable pitch. Figure 12 depicts the multimodal feature fusion heatmap.

Temporal dynamics of multimodal features are used to compare the modalities over time (0–55 s). The three curves shown indicate how emotional change, represented in red, speech intensity, represented in blue, and pitch variability, represented in green, change with respect to time. All three modalities show clear ups and downs that indicate temporal coherence between emotions and the vocal system. The red curve initially has a peak around 5–10 s indicating pleasant emotion, a strong drop around 20–25 s indicating unpleasant emotion, and a subsequent rise again around 35–45 s and a drop around 50 s, displaying that a person shows a transition of emotions over time. The blue curve initially starts with high vocal energy, then drops steadily around 20–25 s, then rises around 35–40 s, and drops again around 45 s. This variation denotes that speech intensity is high during positive emotions and lowest during negative phases. The green curve initially rises sharply, then maintains stable, high changes around 10–35 s and declines at the end. Figure 13 depicts the temporal dynamics of multimodal features.

Multimodal features are visualized using t-SNE visualization that distinguishes the disorders based on the features extracted as shown in Figure 14.

The model performance metrics are evaluated and compared between the unimodal and multimodal model in terms of accuracy, precision, recall, and F1-score. The results prove that the multimodal model performs better than the unimodal model with accuracy around 90%, precision around 85%, recall around 86%, and F1-score around 86%, as shown in Figure 15a, and the ROC curve for the various psychological disorders are depicted in Figure 15b, where ROC for anxiety and PTSD shows AUC of about 0.967, depression of about 0.963, and control of about 0.966, respectively.

Speech signals that are extracted through quantifiable features are evaluated numerically through the Mel-frequency Cepstral Coefficient (MFCCs), which describes the spectral shape of speech that is received from the sound perceived by humans. Figure 16a depicts the MFCC1 difference between control and diseased patients. The schizophrenia patients’ group exhibits higher MFCC1 values where the median is around 1 compared to that of the control group. The features are represented in a Principal Component Analysis (PCA) scatter plot that distinguished control and patient data, shown in Figure 16b.

Physiological data obtained as EEG signals are pre-processed and sent to the EEGNet framework. The model is trained and classified under two classes: healthy (class 0 with 2540 samples) and schizophrenia-affected patients (class 1 with 3089). The dataset contains 14 healthy controls and 14 schizophrenia patients. The EEG data is taken from a 14-channel lightweight system that provides minimal technical training and easy setup that is suitable for taking data in psychiatric scenarios. This 14-channel EEG includes frontal, central, and temporal regions placed at 14 different positions using electrodes. This EEG biomarkers help to capture the features relevant to schizophrenia by tracing the alpha and frontal theta irregularities which causes auditory and temporal responses. This framework also supports the integration of other modalities with enhanced performance. Recent studies have proven results for effective diagnosis of schizophrenia using 8–16 channel systems that ensure critical neural biomarkers are identified without high-density arrays. The dataset comprises 80% as training and 20% as testing data. The training and validation accuracy and loss were evaluated for both control group and schizophrenia patient group. The model achieved a precision of 0.99, a recall of 0.98, and an F1-score of 0.99 for healthy patients and a precision of 0.98, a recall of 0.99, and an F1-score of 0.99 for affected patients. The overall accuracy was achieved as 0.99 for a total of 5629 samples. Taking the macro and weighted accuracy, the ROC AUC was achieved as 0.998. Figure 17a displays the training and validation accuracy for healthy controls, and (b) displays the training and validation loss for healthy controls.

Figure 18a displays the training and validation accuracy for schizophrenia patients, and (b) displays the training and validation loss for schizophrenia patients.

To differentiate the performance between the existing CNN-LSTM method and the proposed EEGNet method, the features extracted from these architectures are analyzed using the t-distributed Stochastic Neighbor Embedding (t-SNE) technique, which is used for visualizing high-dimensional data. This helps to better separate the data points and minimizes overlapping for better classification. This method shows that the EEGNet feature space shows better separation of data points and very minimal overlapping, which distinguishes healthy and affected patients’ classes easily compared to the CNN-LSTM feature space, which has maximum overlapping between healthy and affected data points. Thus, more overlapping makes distinguishing difficult. Figure 19a shows the feature space mapping between healthy and affected data points using EEGNet, and (b) shows the feature space mapping between healthy and schizophrenia patients’ data points using CNN-LSTM, where blue represents healthy controls and red represents affected patients.

The test accuracy comparison between EEGNet and CNN-LSTM reveals that EEGNet achieves an accuracy of 0.99. Furthermore, the ROC-AUC score analysis between the proposed and existing architectures indicates that the AUC of the proposed EEGNet reaches 1.0. Figure 20a,b depict the test accuracy and ROC-AUC score between the EEGNet and CNN-LSTM architectures.

The confusion matrix provides insight into the performance and patterns of EEGNet and CNN-LSTM architectures. This confusion matrix represents the model’s detection against true labels and predicted labels between healthy and schizophrenia patients. Based on the confusion matrix, from a total of 5629 random samples in the data, it was determined that 307 samples of healthy controls were miscategorized as schizophrenia patients in CNN-LSTM, while 21 samples of healthy controls were miscategorized as schizophrenia patients by EEGNet architecture. Figure 21a,b depict the confusion matrix between the EEGNet and CNN-LSTM architectures.

### Strengths and Weaknesses

The main advantage of this system is that integration of EEG data with other modalities like textual, audio, and video is feasible for 14 14-channel EEG acquisition that captures temporal, spatial, and task-evoked potentials. However, the weakness of this system is that the misclassification rates indicate that the model requires better specificity since the sample size is small and this constrains the generalizability of results. This method is more effective in identifying schizophrenia patients than differentiating healthy controls. This result, with the existing literature, shows that EEG data can easily identify schizophrenia patients, whereas healthy controls may overlap with other features that lead to false positives. There is a requirement for multiple performance metrics to confirm the results.

## 5. Limitations and Future Scope

Despite several advantages of using a multimodal framework for diagnosis of schizophrenia, there are certain limitations that have to be addressed. First, the generalizability of the result since the dataset is small, which limits the statistical analysis and model to capture diverse features for the diagnosis of schizophrenia. In the future, data can be collected from larger populations to strengthen the efficiency of diagnosis through a wide range of biomarkers and audio–visual features that enhances the classification performance. Secondly, multiple hardware systems are not used to validate the results and findings. Cross-device testing is essential to validate that the framework can maintain its performance across different scenarios, EEG devices and VR devices. While taking real-time data, participants’ comfort, and VR tolerance, cognitive fatigue should also be addressed. Removing the real-time noise and employing efficient artifact removal method can improve the compatibility of usage. Finally, integration of multimodal data leads to ethical considerations. Ensuring participant privacy and informed consent is important for the deployment of devices in real-time environments. In future, ethical impact assessments should be performed to guide the users with safe and acceptable hardware.

## 6. Conclusions and Future Work

The diagnosis of schizophrenia using multimodal data has shown promising results when fed into the EEGNet framework and has improved the accuracy. Multimodal data, such as text and EEG signals, extract the most spatio-temporal and frequency-based features and enhance the accuracy in diagnosing the severity of symptoms. The performance metrics were also evaluated to check how well the model has been trained on the data and predict the disorder at an early stage. This framework has obtained an accuracy of about 0.99. Hence, EEGNet can be a strong method for creating an EEG-based model for the diagnosis of schizophrenia, since it utilizes small datasets effectively and improves efficiency. In the future, the implementation of EEGNet for real-time datasets and multimodal data like audio and video signals can also be explored. Performance can be improved using appropriate optimization techniques and dimensionality reduction techniques for better classification of diseases and accurate detection of schizophrenia.

## Figures and Tables

**Figure 1 diagnostics-15-03081-f001:**
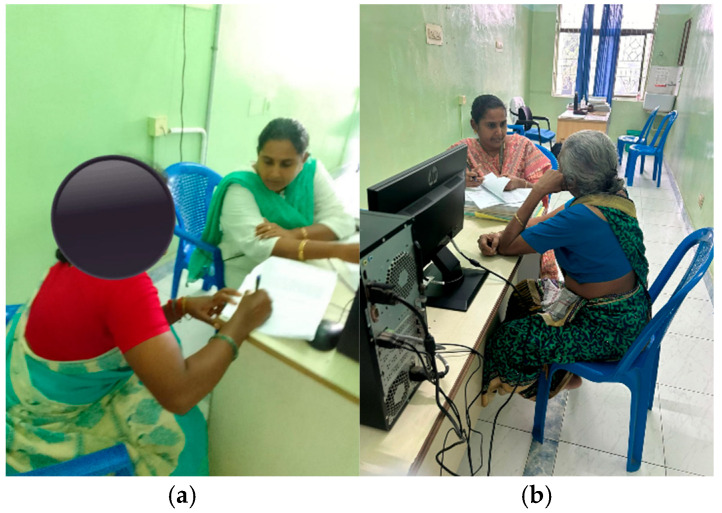
(**a**,**b**) Sample Questionnaire Data Collected from Patients.

**Figure 2 diagnostics-15-03081-f002:**
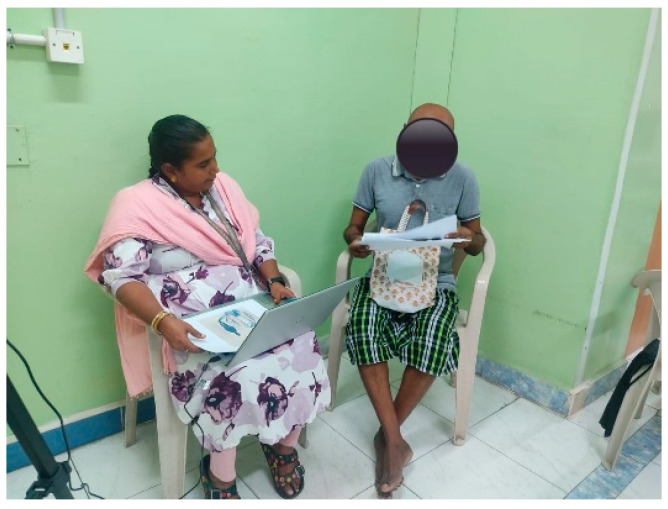
Sample Photo Elicitation Task from Patients.

**Figure 3 diagnostics-15-03081-f003:**
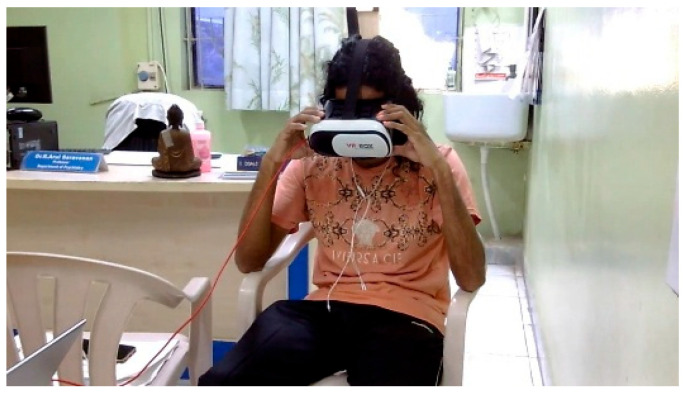
Sample VR Assessment Stimuli from Patients.

**Figure 4 diagnostics-15-03081-f004:**
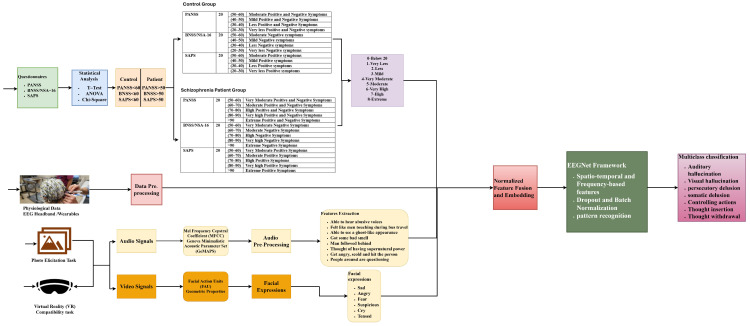
Structure of the Proposed System.

**Figure 5 diagnostics-15-03081-f005:**
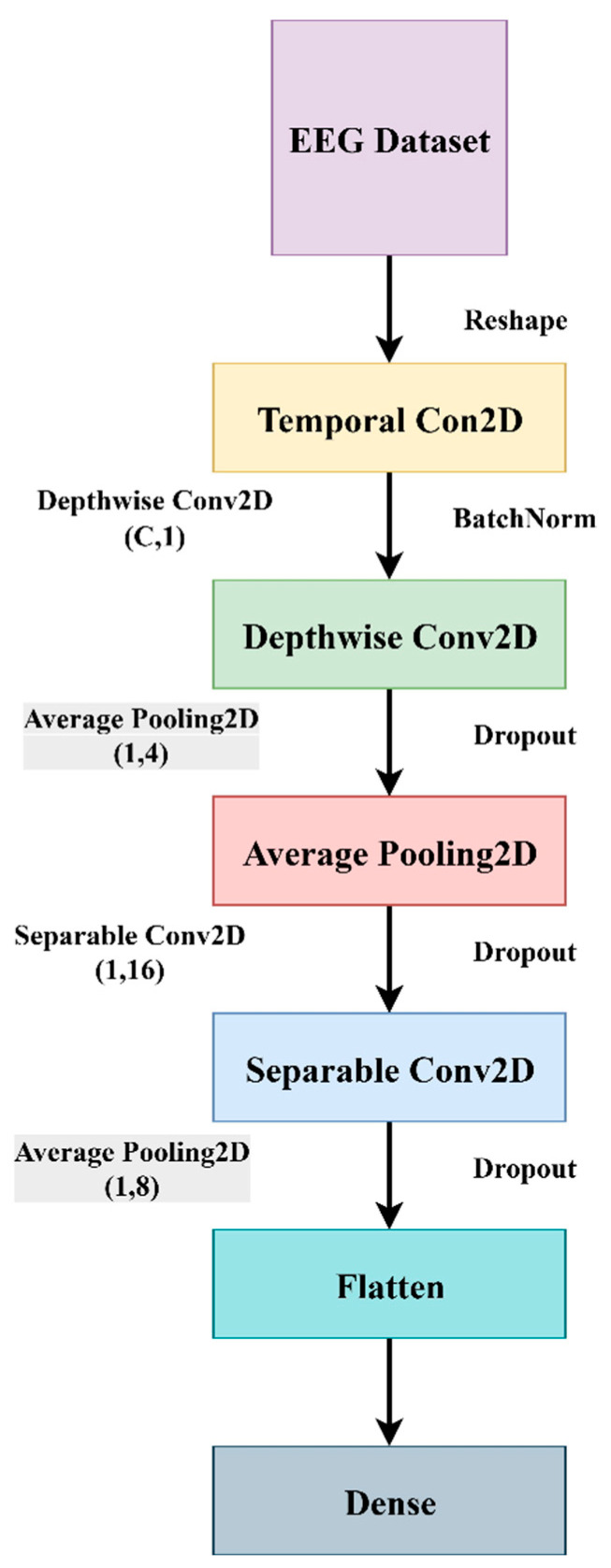
EEGNet Architecture.

**Figure 6 diagnostics-15-03081-f006:**
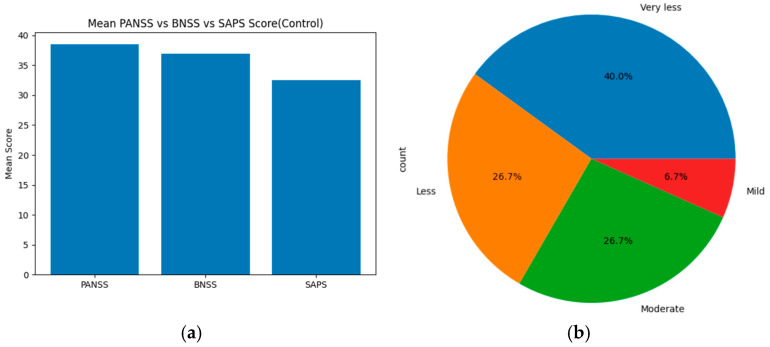
(**a**) Mean score across PANSS, SAPS, and BNSS for Healthy Controls. (**b**) Severity Distribution for Healthy Controls.

**Figure 7 diagnostics-15-03081-f007:**
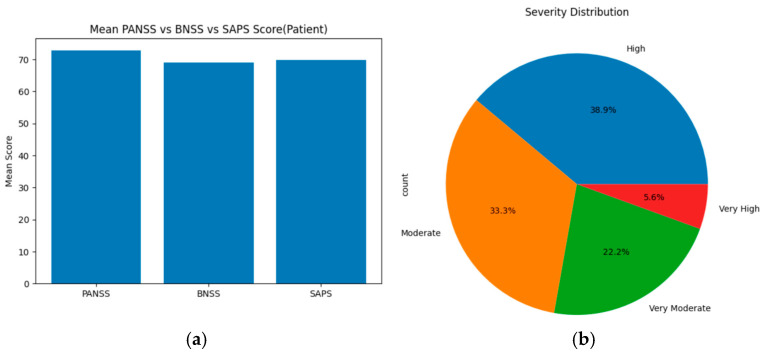
(**a**) Mean score across PANSS, SAPS, and BNSS for schizophrenia patients. (**b**) Severity distribution for schizophrenia patients.

**Figure 8 diagnostics-15-03081-f008:**
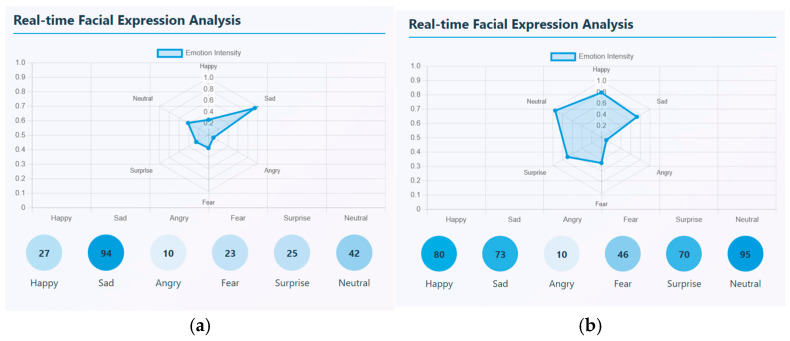
(**a**) Real-time Facial Expression Analysis of a Schizophrenia patient. (**b**) Real-time Facial Expression Analysis of a Healthy Control.

**Figure 9 diagnostics-15-03081-f009:**
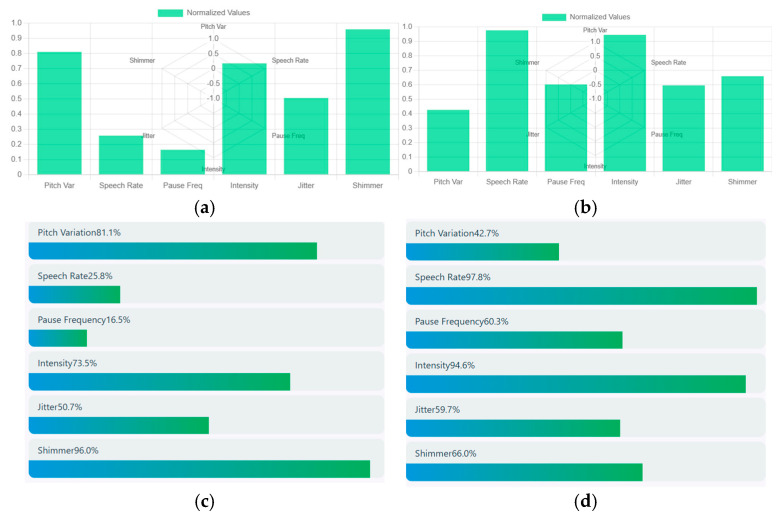
(**a**) Acoustic Speech Features of a Schizophrenia patient. (**b**) Acoustic Speech Features of Healthy Control. (**c**) Comparative Visualization of Acoustic Features of Schizophrenia patient. (**d**) Comparative Visualization of Acoustic Features of Healthy Control.

**Figure 10 diagnostics-15-03081-f010:**
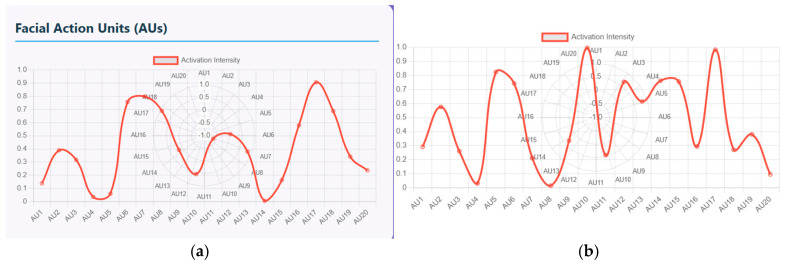
(**a**) Measured Intensity for each AU of a Schizophrenia patient. (**b**) Measured Intensity for each AU of Healthy Control.

**Figure 11 diagnostics-15-03081-f011:**
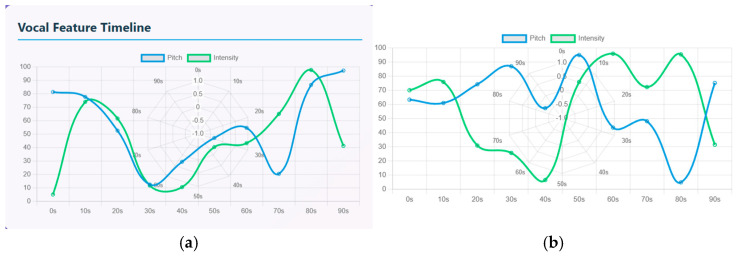
(**a**) Vocal feature timeline of a Schizophrenia patient. (**b**) Vocal feature timeline of a Healthy Control.

**Figure 12 diagnostics-15-03081-f012:**
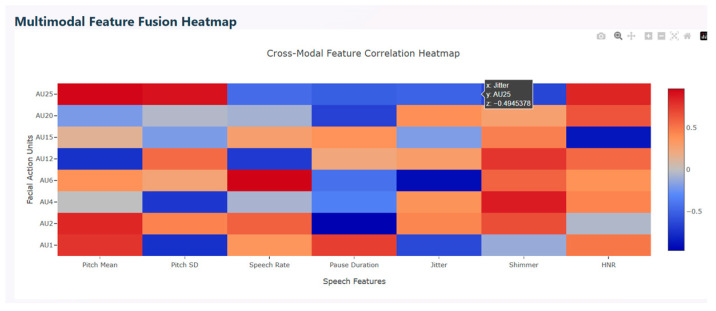
Multimodal Feature Fusion Heatmap.

**Figure 13 diagnostics-15-03081-f013:**
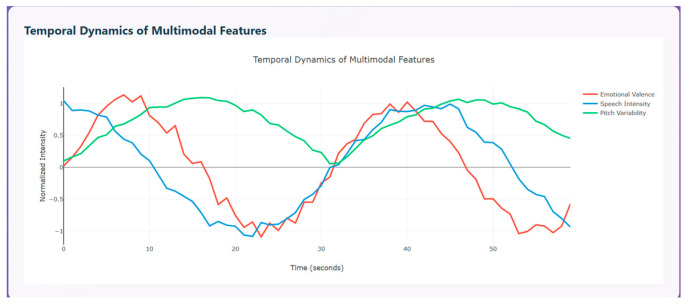
Temporal Dynamics of Multimodal Features.

**Figure 14 diagnostics-15-03081-f014:**
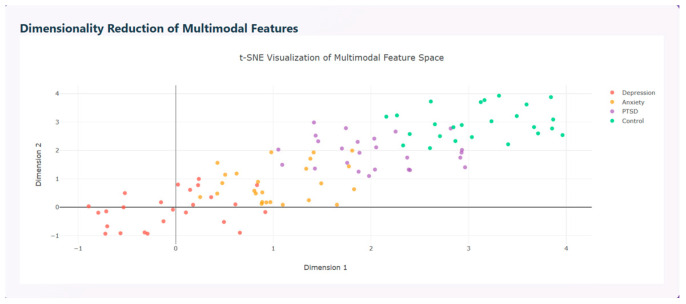
Dimensionality Reduction in Multimodal Features.

**Figure 15 diagnostics-15-03081-f015:**
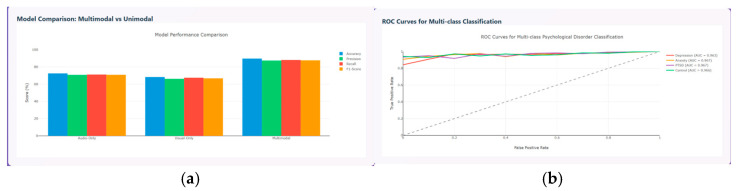
(**a**) Performance Metrics comparison between Unimodal and Multimodal. (**b**) ROC Curves of Multi-class Classification.

**Figure 16 diagnostics-15-03081-f016:**
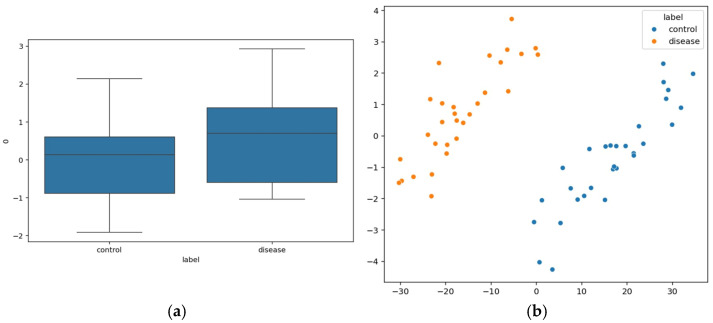
(**a**) MFCC1 differences between Control and Diseased Group. (**b**) PCA Scatter plot of Control and Diseased Group.

**Figure 17 diagnostics-15-03081-f017:**
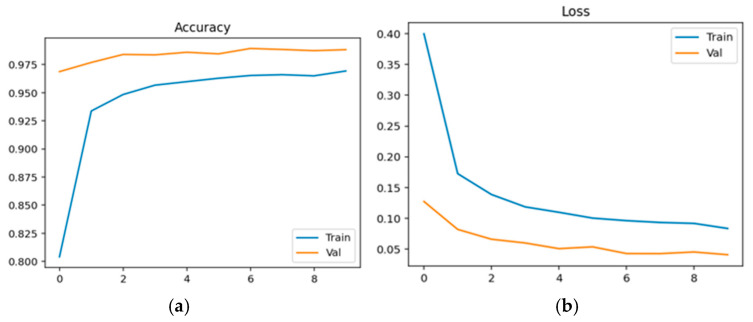
(**a**) Training and Validation Accuracy for Healthy Controls. (**b**) Training and Validation Loss for Healthy Controls.

**Figure 18 diagnostics-15-03081-f018:**
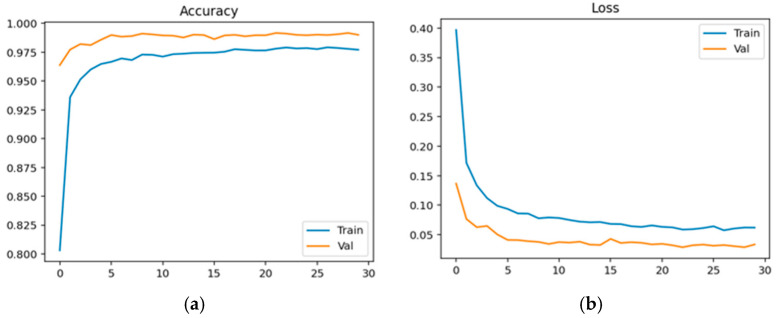
(**a**) Training and Validation Accuracy for Schizophrenia Patients. (**b**) Training and Validation Loss for Schizophrenia Patients.

**Figure 19 diagnostics-15-03081-f019:**
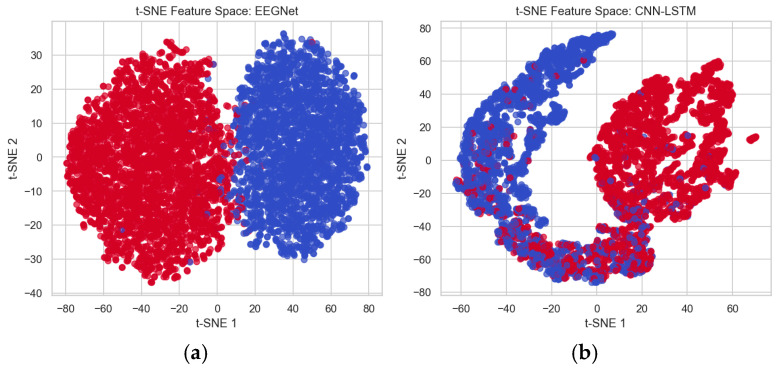
(**a**) t-SNE Feature Space between Healthy (Blue) and Schizophrenia Patients (Red) using EEGNet. (**b**) t-SNE Feature Space between Healthy (Blue) and Schizophrenia Patients (Red) using CNN-LSTM.

**Figure 20 diagnostics-15-03081-f020:**
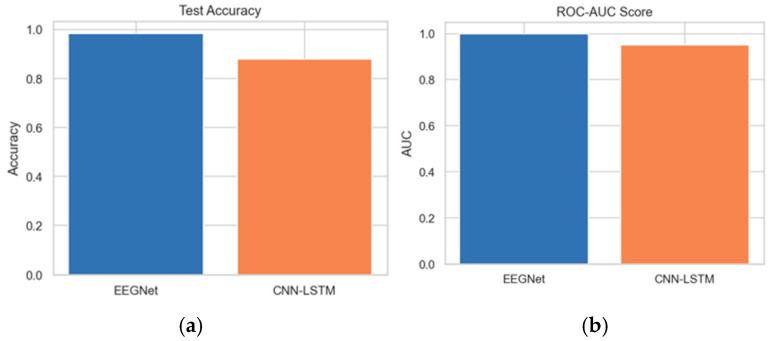
(**a**,**b**) Test Accuracy and ROC-AUC score for EEGNet and CNN-LSTM.

**Figure 21 diagnostics-15-03081-f021:**
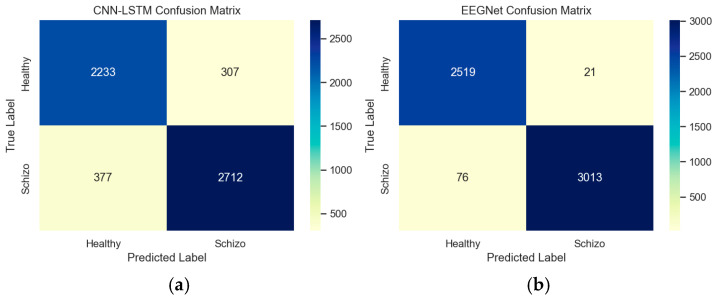
(**a**,**b**) Confusion Matrix between EEGNet and CNN-LSTM.

**Table 1 diagnostics-15-03081-t001:** Comparison between Proposed and Existing Methods.

Parameter	EEGNet	CNN-LSTM
Architecture	Depth-wise separable convolution to separate spatial and frequency filtering Squeeze and Excite attention mechanism Residual connection for better gradient flow and generalization Lightweight and optimized	1D-CNN +LSTM used for temporal modeling No attention mechanism No residual connections Heavier with LSTM blocks
Input and Segmentation	2-s windows with 50% overlap Normalization performed per segment using z-score Detects short abnormal bursts	25-s time frames Z-score and L2 norm, But at the global level Critical temporal events in longer segments
Efficiency and Deployment	A few samples of efficient learning Fewer epochs support real-time inference	More parameters Large batch size and longer training Training LSTMs with More tuning and resources

## Data Availability

The dataset may be obtained by researchers on reasonable request from the corresponding authors.

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
