# Peer review of "Diagnosis of Schizophrenia Using Multimodal Data and Classification Using the EEGNet Framework"

_diagnostics, 2025, doi:10.3390/diagnostics15233081_

Round 1

Reviewer 1 Report

Comments and Suggestions for Authors

Comments on

 Diagnosis of Schizophrenia using Multimodal Data and Classification using the EEGNet Framework

  • At abstract:
  1. The Method section needs to be written in more concise.
  2. This model displays an accuracy 31 of 0.99, precision of 0.99, recall of 0.98, and F1-score of 0.99 for healthy and a precision of 32 0.98, recall of 0.99, and F1-score of 0.99 for schizophrenia affected patients and ROC AUC 33 of about 0.9989. Which model?
  • At section 3.2.

 EEGNet is an efficient and simple CNN architecture that is suitable for EEG-based brain computer interface (BCI) tasks and effectively diagnoses the presence of schizophrenia disorders.

Please declare what the new ideas or modifications added to this existing EEGNET model

  • Please add more recent papers as:
  1. Yasin, S.; Adeel, M.; Draz, U.; Ali, T.; Hijji, M.; Ayaz, M.; Marei, A.M. A CNN-Transformer Fusion Model for Proactive Detection of Schizophrenia Relapse from EEG Signals. Bioengineering202512, 641. https://doi.org/10.3390/bioengineering12060641Please revise the paper for typos.
  2. Supakar, R., Mazumder, S., Neogy, S., Chakrabarti, P., Chakkaravarthy, M. (2024). Detecting Schizophrenia Patients Using Deep Learning Models. In: Mandal, J.K., De, D. (eds) Machine Learning for Social Transformation. EAIT 2024. Lecture Notes in Networks and Systems, vol 1131. Springer, Singapore. https://doi.org/10.1007/978-981-97-7532-3_23
  • Finally, the paper presented a good idea to diagnose Schizophrenia.

Author Response

Reviewer#1, Concern # 1: The Method section needs to be written in a more concise.

Author response:  Thank you for your feedback. We made changes based on your input.

Author action: We updated the method section of the Abstract from page no:1, line no:17 to 25.

Reviewer#1, Concern # 2: This model displays an accuracy 31 of 0.99, precision of 0.99, recall of 0.98, and F1-score of 0.99 for healthy and a precision of 32 0.98, recall of 0.99, and F1-score of 0.99 for schizophrenia affected patients and ROC AUC 33 of about 0.9989. Which model?

Author response:  Thank you for your feedback. We made changes based on your input.

Author action: We updated the manuscript by including the model name, EEGNet model in page no:1, line no:28.

We have contributed by embedding the multimodality features into the EEGNet architecture. We have developed normalized feature set embeddings from four modalities: 1) normalized questionnaire scores  2)normalized speech features extracted by giving Photo Elicitation and VR Assessment Stimuli, and 3) video emotion extracted features by giving Photo Elicitation and VR Assessment Stimuli  4)EEG data features. Those features were concatenated to form a joint feature vector.

Ffusion​=[FEEG​∥Fspeech​∥Fvideo​∥Fq​].

This integration of this multimodal fused feature enables EEGNet to jointly exploit electrophysiological information from EEG and complementary behavioral–affective cues, thereby enhancing classification performance while retaining the architectural efficiency of EEGNet.

There are some key features in the EEGNet architecture that enhance the feature extraction, generalizability and interpretation of schizophrenia detection. The Squeeze and Excite (SE) attention mechanism is used for improving the channel-wise feature responses since this SE block assigns higher weights to important features from EEG channels and suppresses the rest. Thus, the most important features related to brain abnormalities are extracted, which improve interpretability. Another feature is a residual connection that is similar to ResNet architecture, which combines the outputs from the previous layers along with later ones in the neural network, making the system robust. Also, this architecture combines a 2-second overlapping window segmentation that allows detection of even the microstate change, thereby improving the real-time adaptability of the system.

Reviewer#1, Concern # 4: Please add more recent papers as:

  1. Yasin, S.; Adeel, M.; Draz, U.; Ali, T.; Hijji, M.; Ayaz, M.; Marei, A.M. A CNN-Transformer Fusion Model for Proactive Detection of Schizophrenia Relapse from EEG Signals. Bioengineering202512, 641. https://doi.org/10.3390/bioengineering12060641
  2. Supakar, R., Mazumder, S., Neogy, S., Chakrabarti, P., Chakkaravarthy, M. (2024). Detecting Schizophrenia Patients Using Deep Learning Models. In: Mandal, J.K., De, D. (eds) Machine Learning for Social Transformation. EAIT 2024. Lecture Notes in Networks and Systems, vol 1131. Springer, Singapore. https://doi.org/10.1007/978-981-97-7532-3_23.

Author response: Thank you for your feedback. We addressed and updated the inputs. 

Author action: We updated the manuscript and cited the above reference papers in the manuscript on page no: 12.

Reviewer 2 Report

Comments and Suggestions for Authors

The manuscript proposes a practically motivated and clinically relevant real-time multimodal (questionnaires, webcam-based affect capture, VR stimuli, EEG) diagnostic of schizophrenia using an EEGNet framework. The integration of standardized clinical scales, active elicitation tasks, and lightweight neural architecture towards real-world and scalable mental-health support systems is laudable. The reported performance also suggests good classification and can help in early diagnosis in resource-constrained settings.

The authors should take note of several points for better scientific rigor, transparency, and clarity of the study design and presentation. While multimodal data collection is lauded in the abstract and described in the methodology section, the core experiments and reported figures are only based on EEG signals while the audio–visual modalities remain as conceptual information with no quantitative treatment or fusion approach. The manuscript needs to clearly delineate between what has been implemented vs. proposed and what is not included should be reported as limitations. In experimental comparisons, most numbers are coming from a publicly available dataset from previous literature instead of the primary clinical dataset collected for this work; hence the study design should clearly report how much of the reported numbers are from real-patients vs. training/testing on a repository. Very high accuracy values (≈99%) also require better methodological justification and attention for overfitting: (i) what was the class-balancing strategy? (ii) train-validation-test data splitting was not subject-based but within subjects (samples) despite inter-subject variability, how was the bias in generalizability controlled? (iii) is cross-site or leave-one-subject-out evaluation considered? Methodological caveats and justifications should be more detailed, including pre-processing choices (sampling rate, filtering, segmentation parameters), any data augmentation or synthetic data generation strategies, and the selection criteria and labeling strategy.

In several other sections, some statements need better clinical grounding or contextualization in practice. While PANSS, SAPS, BNSS questionnaires are referenced and their use in the study is correct, the statistical results reported in symptom categories and aspects need to be better framed in comparison to established clinical severity thresholds and not just ANOVA p-values in Table 1. The latest consensus updates on early psychosis diagnosis need to be briefly summarized to make the case for the current EEGNet-based biomarker approach as complementing DSM-5/ICD-11 mental status assessments in the introduction.

Feature space comparisons in Fig. 5 and confusion matrices in Fig. 7 would benefit from more detailed legends (number of subjects, channels, total trial numbers) as well as a brief rationale on the selection of a 14-channel acquisition setup since many EEG biomarkers of schizophrenia have focused on frontal temporal functional networks in need of denser arrays.

In the Related Work section, while some reviewed papers discuss the federated learning concept and various medical applications, the majority of citations do not necessarily relate to schizophrenia EEG biomarkers. The paper organization needs to address and summarize more of the recent and relevant multimodal schizophrenia literature to better situate and contextualize the current work’s specific focus on real-time multi-source data fusion and approaches used.

The writing style would also benefit from the removal of narrative reporting in the methodology section and a more stepwise and reproducible protocol description with key choices in the implementation (sampling rate, filtering, segmenting, augmentation choices, labeling strategy, etc.) and clarity on feature dimensions when interpreting the results.

Finally, one paragraph should be added to review recent literature specific to EEG-based schizophrenia biomarkers as well as multimodal information fusion for disease markers to provide better scientific context.https://doi.org/10.1016/j.cmpb.2021.106007 

https://doi.org/10.1016/j.compbiomed.2023.106741 

https://doi.org/10.1016/j.cmpb.2021.106007

We strongly recommend adding a subsection for limitations and future scope that should mention generalizability, small real-patient cohort, missing multimodal fusion in results, lack of cross-institution validation, and use-case considerations for real-world deployment (comfort, VR tolerance, movement artifacts, etc.). Any ethical implications due to behavioral monitoring can also be acknowledged explicitly.

The work has merit, and the revisions addressing the above points will make the study more robust and clinically meaningful.

Author Response

Reviewer#2, Concern # 1: The authors should take note of several points for better scientific rigor, transparency, and clarity of the study design and presentation. While multimodal data collection is lauded in the abstract and described in the methodology section, the core experiments and reported figures are only based on EEG signals while the audio–visual modalities remain as conceptual information with no quantitative treatment or fusion approach. The manuscript needs to clearly delineate between what has been implemented vs. proposed and what is not included should be reported as limitations. In experimental comparisons, most numbers are coming from a publicly available dataset from previous literature instead of the primary clinical dataset collected for this work; hence the study design should clearly report how much of the reported numbers are from real-patients vs. training/testing on a repository. Very high accuracy values (≈99%) also require better methodological justification and attention for overfitting: (i) what was the class-balancing strategy? (ii) train-validation-test data splitting was not subject-based but within subjects (samples) despite inter-subject variability, how was the bias in generalizability controlled? (iii) is cross-site or leave-one-subject-out evaluation considered? Methodological caveats and justifications should be more detailed, including pre-processing choices (sampling rate, filtering, segmentation parameters), any data augmentation or synthetic data generation strategies, and the selection criteria and labeling strategy.

Author response: Thank you for your feedback. We addressed and updated the inputs.

Author action:  We have updated the above-mentioned changes in the entire manuscript by adding experimental data for audio-visual modalities and their quantitative approach of fusion implementation from page no. 10 to 19.

The audio–visual modalities' results are added from pages 15 to 19

 The fusion approaches are also added on page 10 as below :

This article contributed by embedding the multimodality fussed features into the EEGNet architecture . We have developed normalized feature set embeddings from four modalities: 1) normalized questionnaire scores  2)normalized speech features extracted by giving Photo Elicitation and VR Assessment Stimuli, and 3) video emotion extracted features by giving Photo Elicitation and VR Assessment Stimuli  4)EEG data features. Those features were concatenated to form a joint feature vector.

Ffusion=[FEEG∥Fspeech∥Fvideo∥Fq].

This integration of this multimodal fused feature enables EEGNet to jointly exploit electrophysiological information from EEG and complementary behavioral–affective cues, thereby enhancing classification performance while retaining the architectural efficiency of EEGNet.

Reviewer#2, Concern # 2: In several other sections, some statements need better clinical grounding or contextualization in practice. While PANSS, SAPS, BNSS questionnaires are referenced and their use in the study is correct, the statistical results reported in symptom categories and aspects need to be better framed in comparison to established clinical severity thresholds and not just ANOVA p-values in Table 1. The latest consensus updates on early psychosis diagnosis need to be briefly summarized to make the case for the current EEGNet-based biomarker approach as complementing DSM-5/ICD-11 mental status assessments in the introduction.

Author response: Thank you for your feedback. We addressed and updated the inputs.

Author action: we have updated the above changes in the manuscript from page no. 13 to 14, including the clinical severity symptoms and statistical methods.

Reviewer#2, Concern # 3: Feature space comparisons in Fig. 5 and confusion matrices in Fig. 7 would benefit from more detailed legends (number of subjects, channels, total trial numbers) as well as a brief rationale on the selection of a 14-channel acquisition setup since many EEG biomarkers of schizophrenia have focused on frontal temporal functional networks in need of denser arrays.

Author response: Thank you for your feedback. We addressed and updated the inputs.

Author action: we have updated the manuscript with the above-mentioned changes on page no. 21

Reviewer#2, Concern # 4: In the Related Work section, while some reviewed papers discuss the federated learning concept and various medical applications, the majority of citations do not necessarily relate to schizophrenia EEG biomarkers. The paper organization needs to address and summarize more of the recent and relevant multimodal schizophrenia literature to better situate and contextualize the current work’s specific focus on real-time multi-source data fusion and approaches used.

Author response: Thank you for your feedback. We addressed and updated the inputs.

Author action: we have updated the manuscript, including the recent papers related to EEG biomarkers and multimodal framework for diagnosis of schizophrenia from pages no:3 to 6 and 11 to 13.

Reviewer#2, Concern # 5: The writing style would also benefit from the removal of narrative reporting in the methodology section and a more stepwise and reproducible protocol description with key choices in the implementation (sampling rate, filtering, segmenting, augmentation choices, labeling strategy, etc.) and clarity on feature dimensions when interpreting the results.

Author response: Thank you for your feedback. We addressed and updated the inputs.

Author action: we have updated the above-mentioned changes in the entire manuscript.

Reviewer#2, Concern # 5: Finally, one paragraph should be added to review recent literature specific to EEG-based schizophrenia biomarkers as well as multimodal information fusion for disease markers to provide better scientific context.https://doi.org/10.1016/j.cmpb.2021.106007 

https://doi.org/10.1016/j.compbiomed.2023.106741 

https://doi.org/10.1016/j.cmpb.2021.106007

Author response: Thank you for your feedback. We addressed and updated the inputs.

Author action: we have updated the manuscript, adding the above-mentioned references as [28] and [29] on page no:12

Reviewer#2, Concern # 6: We strongly recommend adding a subsection for limitations and future scope that should mention generalizability, small real-patient cohort, missing multimodal fusion in results, lack of cross-institution validation, and use-case considerations for real-world deployment (comfort, VR tolerance, movement artifacts, etc.). Any ethical implications due to behavioral monitoring can also be acknowledged explicitly.

Author response: Thank you for your feedback. We addressed and updated the inputs.

Author action: We have updated the manuscript by adding limitations and a future scope subsection on page no. 22.

Reviewer 3 Report

Comments and Suggestions for Authors

Dear authors, thank you for an interesting article.

In India, the occurrence of schizophrenia was about 1.41% during lifetime and 0.41% currently. 
what does that mean? 

Typically, this disorder affects individuals between the ages of 13 and 24.

there’s no reference and the typical age is around 18 -20 for men, so around the time they finish high school, and later for women.
before the age of 18 it’s called early schizophrenia

You talk about schizophrenia patients and control group but, How did you select patients in your control group, How many people are there in each group, et cetera?

From 320 participants, 32 individuals with early psychosis and 82 individuals with low or high schizotypy were identified
where does that suddenly come

There were 20 responses received online, consisting of 9 male and 11 female
Similarly, there were 20 responses received from the hospital, consisting of 8 male and 12 female patients 

does that mean that they were 20 patients in your study? 

you say that your model is very accurate, but at the same time you state that:

Based on the confusion matrix, it was inferred that 13.18% of healthy controls were miscategorized as schizophrenia patients and 23.32% of healthy controls were misclassified as schizophrenia patients in non-subject-based testing.
That suggests something different.

please mention the strengths and weaknesses of your study. 

Please rewrite your results and discussion because it’s difficult to follow your logic And arguing.

Author Response

Reviewer#3, Concern # 1: In India, the occurrence of schizophrenia was about 1.41% during lifetime and 0.41% currently.

what does that mean?

Author response: Thank you for your feedback. We addressed and updated the inputs.

Author action:  We updated the manuscript by reframing the statement on page no:2. The updated content is as follows:

According to the National Mental Health Survey (NMHS) in India, the occurrence and prevalence of schizophrenia spectrum disorders were examined during 2015 and 2016 through a multistage, stratified, random cluster sampling technique [1]. It was found that the prevalence of schizophrenia-spectrum disorder was about 1.41% in their entire life, and nearly 0.42% people were currently diagnosed with schizophrenia at that moment.

Reviewer#3, Concern # 2: Typically, this disorder affects individuals between the ages of 13 and 24.

there’s no reference and the typical age is around 18 -20 for men, so around the time they finish high school, and later for women. before the age of 18 it’s called early schizophrenia

Author response: Thank you for your feedback. We addressed and updated the inputs.

Author action:  We updated the manuscript by adding the reference [1] and reframing the statement on page no:2. The updated content is as follows:

Participants who show at least one of the symptoms of delusions, hallucinations, or disorganized speech during a period of one month are included in this study. Further, this disorder is diagnosed between late teens and early thirties. In males, it is found earlier between late adolescence and early twenties, and similarly in females, it is found between early twenties and early thirties. Participants who show a history of autism spectrum disorder, communication disorder in childhood, depressive or bipolar disorder are excluded from this study.

Reviewer#3, Concern # 3: You talk about schizophrenia patients and control group but, How did you select patients in your control group, How many people are there in each group, et cetera?

Author response: Thank you for your feedback. There are two groups in our study. The control group is taken from the healthy population who have not received any treatment for the disorder, and the schizophrenia patients’ group is taken from the hospital, who are diagnosed with symptoms and are receiving treatments. In the real-time data, there are about 20 participants in the control group and 20 in the schizophrenia patient group. Similarly, from online data, there are about 14 individuals in the control group and 14 in the schizophrenia patient group. 

Reviewer#3, Concern # 4: From 320 participants, 32 individuals with early psychosis and 82 individuals with low or high schizotypy were identified. where does that suddenly come?

Author response: Thank you for your feedback. The statement is given in the literature review section.The above statement was mentioned in the reference paper [21]. The authors had taken 114 participants to analyze the schizotypy traits and extract speech and language features. 

To avoid confusion, we have rephrased the sentence as below :

From 114 participants taken to the study by the author of [21], 32 individuals with early psychosis and 82 individuals with low or high schizotypy were identified

Reviewer#3, Concern # 5: There were 20 responses received online, consisting of 9 male and 11 female. Similarly, there were 20 responses received from the hospital, consisting of 8 male and 12 female patients. Does that mean that there were 20 patients in your study?

Author response: Thank you for your feedback. The schizophrenia patients' data are taken from the hospital, and the data collection process is still going on. Most of the patients are vulnerable and hesitate to share their illness outside. So, it has been difficult in collecting more samples for more diverse study and investigation. Till now, nearly 25 real-time patents have been collected from the hospital and 14 patients are from online dataset.

Reviewer#3, Concern # 6: You say that your model is very accurate, but at the same time, you state that:

Based on the confusion matrix, it was inferred that 13.18% of healthy controls were miscategorized as schizophrenia patients and 23.32% of healthy controls were misclassified as schizophrenia patients in non-subject-based testing. That suggests something different.

Author response: Thank you for your feedback. We addressed and updated the inputs.

Author action: actually, the confusion matrix is evaluated for 5,629 total random sample data. We have modified the content to avoid confusion as below :

Based on the confusion matrix, it was inferred that 307 samples of healthy controls were miscategorized as schizophrenia patients in the CNN-LSTM architecture from 5,629 total random sample data, and 21 samples of healthy controls were misclassified as schizophrenia patients from 5,629 total random sample data by the EEGNet architecture

Reviewer#3, Concern # 7: Please mention the strengths and weaknesses of your study.

Author response: Thank you for your feedback. We addressed and updated the inputs.

Author action: We have updated the manuscript by adding a strength and weakness subsection on page 22

Reviewer#3, Concern # 8: Please rewrite your results and discussion because it’s difficult to follow your logic and argument.

Author response: Thank you for your feedback. We addressed and updated the inputs.

Author action: New results and discussion contents are added from page no .13 to 19 for PANSS, SAPS, and BNSS questionnaires, Facial Expression from video, Acoustic Speech Features, and Multimodal Features results

Round 2

Reviewer 3 Report

Comments and Suggestions for Authors

No further comments.